# Achieving the Optimal AgO Concentrations to Modulate the Anti-*Trypanosoma cruzi* Activity of Ag-ZnO/AgO Nanocomposites: In Vivo Investigations

**DOI:** 10.3390/pharmaceutics16111415

**Published:** 2024-11-04

**Authors:** José Rodrigues do Carmo Neto, Yarlla Loyane Lira Braga, Pablo Igor Ribeiro Franco, Jordana Fernandes de Oliveira, Rafael Obata Trevisan, Karen Martins Mendes, Milton Adriano Pelli de Oliveira, Mara Rúbia Nunes Celes, Anielle Christine Almeida Silva, Juliana Reis Machado, Marcos Vinícius da Silva

**Affiliations:** 1Department of Bioscience and Technology, Institute of Tropical Pathology and Public Health, Federal University of Goias, Goiânia 74605-050, GO, Brazil; rodriguesnneto@gmail.com (J.R.d.C.N.); yarlla_lira@hotmail.com (Y.L.L.B.); pablo_igor@hotmail.com (P.I.R.F.); jordana.fer@hotmail.com (J.F.d.O.); mapoliv@ufg.br (M.A.P.d.O.); mrubia_celes@ufg.br (M.R.N.C.); 2Department of General Pathology, Federal University of Triângulo Mineiro, Uberaba 38025-180, MG, Brazil; rafaelotrevisan@gmail.com (R.O.T.); karenmartins2314@gmail.com (K.M.M.); 3Laboratório de Novos Materiais Nanoestruturados e Funcionais (LNMIS), Physics Institute, Federal University of Alagoas, Maceió 57072-900, AL, Brazil; aniellechristineas@gmail.com; 4Department of Microbiology, Immunology and Parasitology, Institute of Biological and Natural Sciences of Federal University of Triângulo Mineiro, Uberaba 38025-180, MG, Brazil; marcos.silva@uftm.edu.br

**Keywords:** heart, cytokines, Chagas disease, intestine, nanoparticles

## Abstract

**Background/Objectives**: For the development of new treatments, the acute phase of Chagas disease (CD) in experimental models acts as a filter to screen out potentially effective interventions. Therefore, the aim of this study was to evaluate ZnO nanocrystals and Ag-ZnO/AgO nanocomposites containing different proportions of silver (ZnO:5Ag, ZnO:9Ag and ZnO:11Ag) in an experimental model of the acute phase of CD. **Methods**: C57Bl/6 mice were infected with 1000 forms of the Colombian strain of *T. cruzi*. The treatment was carried out by gavage with 5 mg/kg/d for 7 consecutive days from the first detection of parasitemia. Weight, parasitemia and survival were assessed during treatment and up to the day of euthanasia. After euthanasia, the cardiac and intestinal parasitism, inflammatory infiltrate, collagen deposition and cytokine dosages were analyzed. **Results**: It was observed that the nanocomposites ZnO:9Ag and ZnO:11Ag were the most effective in reducing parasitemia and increasing the survival of the infected animals. However, pure ZnO induced the maintenance of parasitemia and reduced their survival. The ZnO:9Ag and ZnO:11Ag nanocomposites were able to reduce the number of cardiac amastigote nests. In addition, they were responsible for reducing TNF-α and IL-6 in situ. ZnO:9Ag and ZnO:11Ag induced a reduction in the intestinal inflammatory infiltrate and neuronal protection in the myenteric plexus, as well as reducing TNF-α in situ. **Conclusions:** Based on these results, it is suggested that there is an ideal concentration in terms of the proportion of Ag/AgO and ZnO in nanocomposites for use against CD. Thus, ZnO:9Ag or ZnO:11Ag nanomaterials are potential candidates for the development of new biotechnological products for the therapy of CD.

## 1. Introduction

Chagas disease (CD) is a chronic and systemic parasitic condition caused by the microorganism *Trypanosoma cruzi*. This disease, considered an anthropozoonosis, is endemic to Latin America and is the most significant neglected tropical disease in this region. In 2019, it was estimated that approximately eight million people were infected by the protozoan in these areas, resulting in 12,000 to 45,000 deaths annually and around 28,000 new cases [1]. The associated global health costs related to Chagas disease amount to approximately 627.46 million dollars per year. In addition, the economic impact resulting from the premature death of patients and loss of productivity varies between 1 and 2 billion dollars [2,3,4].

There is currently no satisfactory treatment for the chronic phase of CD. The first drug discovered, Nifurtimox, was introduced in 1960. However, due to its high toxicity, its production was gradually discontinued and its commercialization banned in several countries during the 1980s [5,6]. In addition, cases of parasite resistance to this drug have been reported [7]. Benznidazole is therefore the only drug available in most Latin American countries [8]. 

Benznidazole is currently produced by the Pharmaceutical Laboratory of the State of Pernambuco (LAPEFE) in Brazil. Studies suggest that the drug has several mechanisms of action involving changes in the parasite’s DNA, proteins and lipids [9]. Although Benznidazole is effective in the acute phase of the disease, its benefit is limited in the chronic phase [10,11]. It is responsible for inducing side effects such as skin rashes, epigastric pruritus, nausea, abdominal swelling and eosinophilia [11,12]. Due to these factors, plus the long period of drug administration (30 to 60 days), there is low patient adherence to treatment [11,12]. In view of the lack of safer and more effective substitutes for the treatment of CD and the lack of vaccines, new alternatives that are less toxic and more selective for this protozoan are needed. 

Recent advances in nanotechnology open promising possibilities for the discovery of new components in the treatment of CD. Among these advances are metallic nanoparticles (NPs), such as zinc oxide (ZnO). These semiconductor NPs have the ability to absorb and emit ultraviolet light, presenting catalytic properties that are directly related to the generation and induction of reactive oxygen species (ROS), giving them antimicrobial potential [13]. Most studies suggest that ZnO NPs participate in the suppression of the antioxidant system through the generation/induction of ROS (e.g., superoxide anions, hydroxyl radicals and hydrogen peroxide), which leads to a stress-induced oxidative/nitrosative condition [14,15]. As a result, ROS and nitric oxide (NO) damage the proteins, lipids and DNA of normal cells, as well as bacteria, fungi, protozoa and helminths, leading to their death [14,15,16,17,18]. Although the death mechanism is shared in host cells and pathogens, these nanomaterials may have a selective capacity for pathogens, which still needs to be elucidated better [19].

In this context, the aim of this study was to evaluate the in vivo trypanocidal effect of ZnO nanocrystals and Ag-ZnO/AgO nanocomposites in terms of clinical and parasitological parameters and cardiac and intestinal immunopathological aspects in an experimental model of the acute phase of CD. Our results indicate that these nanomaterials could offer a new therapeutic approach to combating this infection caused by *T. cruzi*. 

## 2. Materials and Methods

### 2.1. Nanomaterials

Both pure ZnO nanocrystals and Ag-ZnO/AgO nanocomposites were used. They were prepared using a method under patent application (BR 10 2018 0077147). The structural and morphological properties of the ZnO nanocrystals and Ag-ZnO/AgO nanocomposites were characterized previously and are summarized in Table 1 [19,20]. Here, the Ag-ZnO/AgO nanocomposites are labeled according to the concentration of silver (Ag) in their final structure throughout the text as follows: ZnO, ZnO:5Ag (5% of Ag), ZnO:9Ag (9% of Ag) and ZnO:11Ag (11% of Ag).

### 2.2. Animals

Male C57Bl/6 mice, aged 8–10 weeks old and weighing 22–27 g, were employed as the experimental subjects. The animals were housed in the controlled environment of the Centro Multiusuário de Pesquisa e Experimentação Animal at the Institute of Tropical Pathology and Public Health. Specifically, they were accommodated in polypropylene cages measuring 414 mm × 168 mm, maintaining a temperature range of 20 to 25 °C, humidity levels between 45 and 55% and a continuous supply of fresh air. The mice were provided with nutritionally characterized food (Nuvilab-CR1, NUVITAL, Paraná, Brazil) and had ad libitum access to water. All components of their housing, including the water, wood shavings and feed, underwent autoclaving before their utilization to ensure their sterility. Ethical considerations were meticulously addressed, with the study protocol (No. 091/20) receiving approval from the Ethics Committee for the Use of Animals at the Federal University of Goiás. The research team implemented stringent monitoring practices, conducting daily observations of the animals’ behavior to promptly identify humane endpoints if they were warranted.

### 2.3. Experimental Design: Infection, Treatment, Euthanasia and Organ Harvesting

This study was performed using three major experimental groups (Table 2): uninfected and untreated mice (healthy control—NF), infected and untreated mice (infected control/vehicle—INF) and infected and treated mice (the intervention group). In addition, the treated group was subdivided according to the intervention used: treatment with ZnO, ZnO:5Ag, ZnO:9Ag or ZnO:11Ag. 

For infection, blood trypomastigotes previously maintained in Balb/c mice were inoculated into the C57Bl/6 mice subcutaneously (1 × 10^3^ blood trypomastigotes) [21,22]. The concentration used for the treatment, regardless of the type of nanomaterial, was 5 mg/kg for seven consecutive days after the detection of parasitemia, given via gavage in PBS (100 µL).

All of the groups were euthanized after 30 days of infection/maintenance. This was carried out by cervical dislocation (after sedation with a solution of 5% xylazine hydrochloride and 10% ketamine intraperitoneally), followed by arterial blood collection and necropsy through a dorsolateral incision. After the groups were euthanized, two fragments of their hearts (the apex and base) and intestines (the proximal and distal colon) were collected. The base of the hearts and the distal fragments of the intestines were washed in PBS and then processed and stained for histopathological analysis. The apex of the hearts and the proximal fragments of the intestines were transferred into 2 mL centrifuge tubes and stored in a −80 °C freezer for subsequent cytokine quantification. The complete experimental design is depicted in Figure 1.

### 2.4. Monitoring Parasitemia, Weight and Survival

The parasitemia of the treated and untreated infected mice was monitored from the 5th day after infection to the 29th day after infection. Five μL of blood was taken from the mice through their tail veins to assess the level of parasitemia. The parasites were counted in 50 random fields under a slide and a coverslip using an optical microscope with 400× magnification [21,22,23]. The parasites were observed every 4 days after the first detection of parasites in the animals’ blood. The result on parasitemia was normalized by the area of the slide and the area observed under the microscope according to the magnification used [23]. The weight of the animals was recorded from the first detection of circulating parasites and then successively every four days. In addition, on the day of euthanasia, the heart of each mouse was weighed. The mice were observed daily to monitor their survival.

### 2.5. Processing of the Samples for Histopathology

The heart basal fragment and the intestinal distal fragment of each mouse destined for histology were fixed in 4% paraformaldehyde for a maximum of 48 h. Next, the fragments were transferred into an ascending series of ethyl alcohols for dehydration and then diaphanized in xylene and embedded into paraffin. The fragments were oriented along the longitudinal axis perpendicular to the microtomy plane to obtain cross-sections. For each organ, three sections were made in series, 25 µm apart. These sections were adhered to slides with polylysine adhesive and dried. Routine staining was then carried out for each specific evaluation of the work, which will be explained below [24]. 

#### 2.5.1. Cardiac Tissue Parasitism

Cardiac tissue parasitism was measured by counting the number of amastigote nests in the ventricles of the hearts on Hematoxylin–Eosin (HE)-stained slides. Tissue parasitism was assessed in 20 random fields of the heart and in three serial sections (total n = 60 fields) using a 40× objective on a common optical microscope (Axiolab—Zeiss, Jena, Germany) [25]. The result was given as parasites/field and was obtained using the following formula:Parasites/field = Average number of amastigote nests in the three fragments/20

#### 2.5.2. Quantification of Cardiac and Intestinal Inflammatory Infiltrate

The HE-stained slides were used to quantify the cardiac and intestinal inflammatory infiltrate/leukocyte infiltration. The analysis was carried out using a 20× objective on a common optical microscope (Axiolab—Zeiss, Germany). For both organs, 10 fields were evaluated at random. For the heart, the ventricles were evaluated in three serial sections (n = 30 fields/mouse). For the intestine, the analysis focused on the submucosal and muscular bed, also in three serial sections (n = 30 fields/mouse). The intensity of the inflammatory process was assessed and classified as 0 (normal), 1 (mild), 2 (moderate) or 3 (severe) according to the field assessed. After categorizing the fields, the average of each case obtained was classified according to the following score: 0–0.3 = normal; 0.4–1.0 = mild; 1.1–2 = moderate; and 2.1–3 = severe (adapted from [24,26,27]). The result was given as a score using the following formula:Inflammatory infiltrate (score) = ∑ intensity of the inflammatory process in the 30 fields/30

#### 2.5.3. Quantification of PGP 9.5 Immunoreactivity in the Myenteric Plexus

Immunohistochemistry for PGP 9.5 (Thermo Scientific™, Wilmington, DE, USA, catalog #38-1000) was carried out in the intestines of the infected mice and controls, whether treated or untreated. After blocking endogenous peroxidase through treatment with H_2_O_2_ in methanol, the slides were washed and incubated with PBS containing 2% BSA. Each section was then incubated with the primary antibody at a concentration of 1:200 and incubated overnight in a refrigerator. The slides were then washed in PBS and incubated again for 2 h with the detection kit (LSAB—DAKO Corporation, Carpinteria, CA, USA). The slides were washed and developed with DAB (1,4-dideoxy-1,4-imino-D-arabinitol-Diaminobenzidine—DAKO Corporation, Carpinteria, CA, USA) for 2 min. The reaction was stopped by washing the slides under running water. The sections were counterstained with hematoxylin and mounted for analysis under an ordinary light microscope. The analysis was based on the evaluation of 10 random fields using a 20× objective on a common optical microscope (Axiolab—Zeiss, Germany) (adapted from [28]). Quantification of the marked area was carried out using AxionCam ICc5 (Zeiss, Germany).

#### 2.5.4. Quantification of Cardiac and Intestinal Collagen

Picrosirius-stained slides were used to morphometrically evaluate the deposition of cardiac connective tissue in the heart and intestine. Analyses were carried out under a microscope using polarized light with a 20× objective and quantification using AxionCam ICc5 (Zeiss, Germany). A total of 10 fields were randomly evaluated in the heart ventricles and intestines (mucosa, submucosa and muscle). The results were expressed as % collagen/animal [24,25].

### 2.6. Obtaining Cardiac and Intestinal Homogenates and Immunological Analysis

On the day of euthanasia, the cardiac apex and the proximal intestinal fragment were transferred into an Eppendorf tube containing a previously identified PBS solution, at a pH of 7.2, along with Complete™ protease inhibitor (SIGMA, St. Louis, MO, USA), and then stored at −80 °C. The fragments were then added into a homogenizer (Dremel, Mount Prospect, IL, USA). The homogenates obtained were centrifuged at 12,000× *g* for 30 min and the supernatant stored at −80 °C for quantification of the cytokines and total proteins.

The cytokines IL-10, TNF-α, IFN-γ, IL-6 and IL-4 present in the cardiac and intestinal homogenates were simultaneously quantified using the Cytometric Bead Array (CBA) technique (Mouse Th1/Th2/Th17 CBA Kit—BD™ Cytometric Bead Array-CBA, San Jose, CA, USA) according to the manufacturer’s protocol. The cytokine-bound beads were then resuspended in 200 µL of wash buffer and transferred into cytometry tubes for acquisition, which was carried out on the same day using the FACSCalibur BD flow cytometer (BD Biosciences, San Jose, CA, USA). The acquisition was analyzed using FCAP ArrayTM program version 2.0 (BD Biosciences, San Jose, CA, USA), and the cytokine concentration was estimated using a linear regression analysis, with the fluorescence obtained from the standard curve of each cytokine and expressed in pg/mL. For the results, the value obtained for each cytokine was normalized to the g of total protein present in the heart and the intestinal tissue. Total proteins were quantified on NanoDrop™ 2000/2000c spectrophotometers (Thermo Scientific™, Wilmington, DE, USA).

### 2.7. Statistical Analysis

The database was set up in the Excel 2016 program, and the statistical analyses were carried out in GraphPad Prism 8.0.1 (Prism 8.0.1, GraphPad Software, San Diego, CA, USA). Verification of the normal distribution of the quantitative variables was assessed using the Shapiro–Wilk test. For comparisons between two groups, the unpaired t-test was used for data with a normal distribution, and the Mann–Whitney test was used for data with a non-normal distribution. For comparisons between more than two groups, the two-way ANOVA test was used for data with a normal distribution, and Tukey’s test was used for data with a non-normal distribution. The Kaplan–Meier analysis was used to construct the survival curve. The results were considered statistically significant when *p* < 0.05.

## 3. Results

### 3.1. A Reduction in Parasitemia and An Increase in Animal Survival Are Dependent on the Concentration of Ag in the Nanocomposites

In order to assess the impact of the nanomaterials on acute *T. cruzi* infection, clinical and parasitological parameters such as weight change (Figure 2A), parasitemia (Figure 2B), relative heart weight (Figure 2C) and survival (Figure 2D) were analyzed. The C57Bl/6 mice were treated for 7 consecutive days with each nanomaterial at a concentration of 5 mg/kg/d via gavage, with treatment starting 9 days post-infection (dpi)—on the first day of detection of circulating parasites—and ending 15 dpi. *T. cruzi* infection induced weight loss from the 25th dpi (*p* < 0.0015) to the 30th dpi (*p* < 0.0001), regardless of treatment, when compared to the healthy group. The group treated with the ZnO nanocrystals also demonstrated a reduction in weight compared to the infected group treated with the vehicle from the 17th dpi to the 30th dpi (PBS 1×) (*p* = 0.0412).

Regarding parasitemia (Figure 2B), the impact of the nanomaterials on a reduction in circulating parasites was observed from the 21st day of infection for the ZnO:5Ag (*p* = 0.0061), ZnO:9Ag (*p* = 0.0087) and ZnO:11Ag (*p* = 0.0021) nanocomposites when compared to the vehicle-treated group. This reduction was maintained over the consecutive days of parasitemia only for the groups treated with ZnO:9Ag (25 dpi, *p* = 0.0004; 29 dpi, *p* < 0.0001) and ZnO:11Ag (25 dpi, *p* < 0.0001; 29 dpi, *p* < 0.0001).

Regarding relative heart weight (Figure 2C), infection, regardless of treatment, induced an increase in this parameter compared to that in the healthy group (*p* < 0.0022). Regarding survival (Figure 2D), it was shown that the higher the concentration of Ag, the greater the animals’ survival from infection, with the groups treated with ZnO:9Ag and ZnO:11Ag having the highest survival percentages (83.33% and 91.67%, respectively). The infected group treated with the vehicle showed 75% survival, followed by ZnO:5Ag with 66.67% and ZnO with 50%. 

### 3.2. Cardiac Histopathological Parameters: The Reduction in Cardiac Parasitism Is Dependent on the Concentration of Ag in the Nanocomposites

Since nanocomposites with a higher concentration of silver (ZnO:9Ag and ZnO:11Ag) reduced parasitemia and increased the survival of infected animals, it was examined whether the interventions would also have an impact on histopathological aspects resulting from experimental Chagas disease. Thus, parameters such as tissue parasitism, inflammation and cardiac collagen deposition were evaluated.

To evaluate cardiac parasitism (Figure 3A), the number of amastigote nests was counted serially in the ventricles of the heart. The pattern observed previously was maintained for this parameter, with a reduction in the number of amastigote nests/field only in the groups treated with ZnO:9Ag (*p* = 0.0190) and ZnO:11Ag (*p* = 0.0429) when compared to the vehicle-treated group. Interestingly, the insertion of silver oxide also reduced the number of amastigote nests, which was significantly demonstrated between the groups treated with ZnO:9Ag (*p* = 0.0412) and ZnO:11Ag (*p* = 0.0385) when compared to the group treated with ZnO nanocrystals.

Regarding the inflammatory infiltrate (Figure 3B), only the ZnO treatment (*p* = 0.0126) resulted in an elevated presence of inflammatory cells in the heart compared to the infected control. The other nanomaterials showed no significant impact on this parameter when compared to the untreated infected group. Notably, the nanocomposites ZnO:5Ag (*p* = 0.0190) and ZnO:9Ag (*p* = 0.0176) demonstrated a reduction in the amount of inflammatory infiltrate compared to that in the group treated with ZnO nanocrystals.

As for collagen deposition (Figure 3C), infection, regardless of treatment, increased the deposition of these fibers in the cardiac tissue (*p* < 0.0226). However, treatment with the nanomaterials was unable to reduce or increase cardiac collagen deposition when compared to that in the vehicle-treated group (*p* > 0.2628).

### 3.3. Intestinal Histopathological Parameters: ZnO:9 and ZnO:11 Nanocomposites Reduce Intestinal Inflammatory Infiltrate and Induce Neuroprotection

In order to assess the impact of the nanomaterials on intestinal histopathological changes induced by the infection, parameters such as the inflammatory infiltrate (Figure 4), the amount of immunoreactive neurons for PGP 9.5 (Figure 5) and collagen deposition (Figure 6) were evaluated. Only the treatments with ZnO:9Ag (*p* = 0.0290) and ZnO:11Ag (*p* = 0.0384) reduced the amount of intestinal inflammatory infiltrate compared to that in the infected group treated with the vehicle (Figure 4A). In the untreated infected group, 85.71% of the animals were categorized as having a moderate inflammatory infiltrate. For the ZnO:9Ag group, 75% had moderate infiltrates and 25% mild. For the group treated with ZnO:11Ag, 77.77% were classified as having a moderate inflammatory intestinal infiltrate and 22.22% as mild (Figure 4B–G).

Immunolabeling by PGP 9.5 was used to determine the neuron density and estimate neuronal destruction (low PGP 9.5/µm2) (Figure 5A). Infection was shown to reduce the amount of labeling for these cells in the intestinal myenteric plexus when compared to that in the healthy group (*p* = 0.0002). In addition, the treatments with ZnO and ZnO:5Ag maintained this pattern, with a reduction compared to that in the healthy group (*p* = 0.0002 and *p* = 0.0061, respectively) and no statistical difference compared to the infected group treated with the vehicle (*p* = 0.8301 and *p* = 0.2593, respectively). Treatment with the ZnO:9Ag or ZnO:11Ag nanocomposites induced preservation of PGP 9.5 labeling, with a significant increase compared to that in the vehicle-treated group (*p* < 0.0424) and the ZnO-nanocrystal-treated group (*p* < 0.0381). In addition, in these two groups, the PGP 9.5 marking was normalized to that found in the healthy group (*p* > 0.2303).

For intestinal collagen deposition (Figure 6A), infection induced an increase in fibers in the organs when compared to the amount in the non-infected group (*p* = 0.0057). This pattern was maintained even with the ZnO (*p* = 0.0235), ZnO:5Ag (*p* = 0.0491) and ZnO:9Ag (*p* = 0.0343) treatments, which did not reduce the amount of collagen when compared to that in the infected group treated with the vehicle (ZnO, *p* = 0.2462; ZnO:5Ag, *p* = 0.7605; ZnO:9Ag, *p* = 0.1379). Only treatment with the ZnO:11Ag nanocomposite was able to reduce collagen deposition, both in comparison with the infected group (*p* = 0.0291) and the ZnO-nanocrystal-treated group (*p* = 0.0325), with no significant difference from the non-infected group (*p* = 0.2608). The group treated with ZnO:9Ag showed a trend towards reduced collagen deposition compared to that in the group treated with the ZnO nanocrystals (*p* = 0.0591).

### 3.4. Treatment with ZnO:9 and ZnO:11 Nanocomposites Reduces Inflammatory Cytokines in the Heart and Intestine

In order to assess the impact of the nanomaterials on the amount of cardiac cytokines and how this could impact the infectious process, IFN-γ, TNF-α, IL-6, IL-4 and IL-10 were evaluated (Figure 7). For TNF-α (Figure 7A), infection was also able to induce an increase in this cytokine in all groups when compared to the level in the healthy control (*p* < 0.0079). However, only the treatments with ZnO:9Ag (*p* = 0.0379) or ZnO:11Ag (*p* = 0.0186) reduced the amount of TNF-α when compared to that in the infected group treated with the vehicle. The infection was able to induce an increase in IFN-γ (Figure 7B) in the heart tissue compared to the level in the healthy group in all groups (*p* < 0.0016). However, the treatments did not induce any difference in the amount of this cytokine compared to that in the infected group treated with the vehicle (*p* > 0.2222).

Infection induced no statistical difference in the production of IL-4 (Figure 7C) or IL-10 (Figure 7D) when compared to that in the healthy group (IL-4: *p* > 0.2547; IL-10: *p* > 0.4021). Following the same pattern, no treatment was able to induce a significant change in the levels of these cytokines compared to those in the infected group treated with the vehicle (IL-4: *p* > 0.6671; IL-10: *p* > 0.3723). For IL-6 (Figure 7E), infection induced an increase in this cytokine in all groups compared to the amount in the healthy control (*p* < 0.0043). Similar to what was found for TNF-α, only the ZnO:9Ag (*p* = 0.0360) and ZnO:11Ag (*p* = 0.0360) nanomaterials induced a reduction in IL-6 compared to the amount in the infected group treated with the vehicle.

In order to assess the impact of the nanomaterials on the amount of intestinal cytokines and how this could impact the infectious process, IFN-γ, TNF-α, IL-6, IL-4 and IL-10 were evaluated (Figure 8). For TNF-α (Figure 8A), infection was able to induce an increase in this cytokine in all groups when compared to the level in the healthy control (*p* < 0.0353). However, only the treatments with ZnO:9Ag (*p* < 0.0442) or ZnO:11Ag (*p* < 0.0242) reduced the amount of TNF-α when compared to that in the infected group treated with the vehicle and the group treated with ZnO nanocrystals. Infection was also able to induce an increase in intestinal IFN-γ (Figure 8B) compared to that in the healthy group in all groups (*p* < 0.0293). However, the treatments did not induce any difference in the amount of this cytokine compared to that in the infected group treated with the vehicle (*p* > 0.4079) or the pure nanocrystals (*p* > 0.1812).

With regard to Th2 and Treg cytokines, infection did not induce a statistical difference in the production of IL-4 (Figure 8C) or IL-10 (Figure 8D) when compared to that in the healthy group (IL-4: *p* > 0.1022; IL-10: *p* > 0.3939). A reduction in IL-4 was only observed in the ZnO:9Ag-treated group when compared to the ZnO-nanocrystal-treated group (*p* = 0.0360). For IL-6 (Figure 8E), infection induced an increase in this cytokine in all groups compared to the healthy control (*p* < 0.0368). Furthermore, the use of silver-doped nanomaterials reduced the levels of this cytokine when compared to those when using the ZnO nanocrystals (*p* < 0.0496).

## 4. Discussion

There is a great need to develop alternative and more effective treatments for Chagas disease since the current treatments exhibit high toxicity and low adherence and efficacy depending on the stage of the disease and the characteristics of the protozoan. The aim of this study was to evaluate the potential of ZnO nanocrystals and ZnO/AgO nanocomposites containing different concentrations of Ag in acute experimental Chagas disease. In fact, the amount of Ag in the nanomaterial formulation was directly related to the trypanocidal profile, with reduced parasitemia, increased survival and altered cardiac and intestinal immune response (ZnO:9Ag and ZnO:11Ag; results summarized in Table 3).

The use of zinc-oxide-based nanomaterials has been widely described in the literature with various functionalities, especially microbicidal and immunomodulatory. However, before checking their potential against infectious diseases, it is necessary to understand the toxicity of nanomaterials and how the nanoformulations impact this parameter. Our group therefore tested different forms of ZnO NPs to assess their toxicity. It was found that the ZnO nanocrystals were less toxic than amorphous nanoparticles of the same material [29]. Next, to enhance the characteristics of the ZnO nanocrystals, they were combined with different amounts of AgO, resulting in ZnO/AgO nanocomposites, named ZnO:5Ag (51% Ag-ZnO and 49% AgO), ZnO:9Ag (35% Ag-ZnO and 65% AgO) and ZnO:11Ag (32% Ag-ZnO and 68% AgO) here. All these materials were previously characterized using data published in other works by this group [19].

That said, this study showed that the ZnO:9Ag and ZnO:11Ag nanocomposites, although they did not alter the weight loss of the infected animals, did reduce parasitemia and the number of amastigote nests in their heart tissue, regardless of the intensity of the inflammatory infiltrate. For the intestine, it was shown that these same treatments were able to reduce the amount of intestinal inflammatory infiltrate, induce neuroprotection in the myenteric plexus and ultimately increase the survival of the animals when compared to that in the infected group treated with ZnO nanocrystals. This result suggests that a reduction in ZnO in the mass of the nanomaterial and an increase in AgO in the structure improved the biological effect of the nanocomposite against this protozoan in an in vivo model, which marked it as a potential candidate for chronic-phase tests. In vitro, this pattern was also demonstrated for Leishmania braziliensis, with nanomaterials with a higher amount of AgO and a lower amount of ZnO (ZnO:9Ag and ZnO:11Ag) resulting in greater toxicity to the parasite (lower IC50) [19].

In another context, AgO nanoparticles have also been reported to have a better microbicidal effect on bacteria when compared to ZnO under the same conditions [30]. By combining these two metals, it is possible that the phenomena behind the cytotoxicity of each are also combined, which is not found when the nanocrystals are used alone. In addition, it can be suggested that there are more appropriate proportions of zinc oxide and silver to use to enhance the trypanocidal profile, since the ZnO:5Ag nanocomposite did not have a significant effect on improving cardiac and intestinal histopathology.

When administered orally, ZnO nanoparticles are rapidly dissolved in the acidic pH of the stomach (at a pH of 1.5–2), releasing Zn^2+^ ions, which are absorbed into the systemic circulation [31,32]. In addition, studies with different experimental models suggest that the absorption of these nanoparticles is mostly in their ionic form and partially in their particle form, with a difference in the absorption depending on the intestinal tract [33,34,35,36,37]. The pharmacokinetic parameters described for this type of nanomaterial vary depending on the size, concentration, form of production, route of administration and experimental model used. Thus, due to the lack of standardization between studies, it is complex to establish a standard for these parameters. For Ag and ZnO NPs, some of their mechanisms are related to cell damage: firstly, direct contact with cells (a direct cytotoxic effect), the dissolution of ions (Ag^+^ or Zn^2+^) and, consequently, the induction of oxidative/nitrosative stress [38,39,40,41,42,43,44]. Thus, it is believed that the combination of oxides in the ZnO:9Ag and ZnO:11Ag nanocomposites added to the cytotoxic mechanisms of the nanomaterials and therefore had an impact on the parasitological parameters in the animals.

Regarding the immunological parameters, the results show that the treatments with the ZnO:9Ag or ZnO:11Ag nanocomposites induced a reduction in pro-inflammatory components, both in the heart (TNF-α, IL-6) and in the intestine (TNF-α). In contrast, the use of the ZnO nanocrystals induced and maintained a high pro-inflammatory profile in the organs. In fact, *T. cruzi* infection, especially in the acute phase, involves a complex level of cytokines and inflammatory cells in the different tissues affected. TNF-α, IL-6 and IFN-γ are classic cytokines involved in activating the initial immune response against this protozoan and thus modulate the response in favor of parasite control. However, high and persistent levels of these cytokines contribute to tissue damage, which favors the detriment of the host and not just the parasite.

TNF-α has already been reported to be involved in various processes that harm the host. In fact, the persistent production of this cytokine is suggested as a participant in the pathogenic process of Chagas cardiomyopathy and is associated with a worse prognosis from the acute phase onwards. This phenomenon occurs due to the persistent induction of reactive nitrogen and oxygen species, which accumulate intracellularly and jointly induce mitochondrial dysfunction [45] and cardiac function disorders [46], for example. In addition, the more virulent the strain of *T. cruzi*, the higher the levels of TNF-α produced, with consequent exacerbation of pro-inflammatory cytokines, which can contribute to animal death [47]. Even so, in an animal model, (from 15 to 120 days after infection) TNF-α production was shown to be persistent in the heart throughout infection [48]. Thus, the increased ratio of TNF-α to IL-4 and IL-10 in the mice treated with the ZnO nanocrystals may have been related to the increased mortality in this group. Meanwhile, the opposite was observed in the groups treated with the ZnO:9Ag and ZnO:11Ag nanocomposites, with a reduction in this cytokine and an increase in survival.

TNF-α participates in the process of cachexia, a marker of systemic inflammation that is associated with weight loss in animals [49]. In fact, for *T. cruzi* infection, it has been shown that blocking TNF-α in the early stages of the disease, and not IFN-γ or IL-6, results in reduced weight loss in animals in the acute phase [49]. The opposite was shown for IFN-γ, and when this cytokine was blocked, the animals’ weight loss intensified, suggesting a protective role in this context [49]. A more recent study showed that other factors may be associated with this weight loss in addition to the involvement of TNF-α, by interrupting the catabolic and anabolic metabolism of adipocytes, tipping the scales towards a state similar to adipose tissue atrophy and the acquisition of an inflammatory phenotype [50].

High levels of IL-6 have also been linked to the severity of Chagas disease in both humans and experimental models, with the development of cardiac hypertrophy, arrhythmias and cardiac fibrosis [51]. In addition, high levels of IL-6 are also associated with hemostatic dysfunction. It is therefore suggested that IL-6 is an important marker of cardiac tissue damage and systemic vascular inflammation in Chagas disease with prognostic value [52,53,54,55]. Interestingly, one of the main components capable of inducing an increase in IL-6 is *T. cruzi* trans-sialidase [56], which suggests an important presence of the parasite in inducing the signaling of this cytokine.

Thus, the reduction in TNF-α and IL-6 found in the heart tissue and TNF-α in the intestine can be attributed to two mechanisms: (1) the reduction in parasitemia and amastigote nests caused by the treatment with ZnO:9Ag or ZnO:11Ag being associated with a reduction in trans-sialidase; a reduction in stimuli for the production of these cytokines; and consequently lower levels in the heart tissue and (2) immunomodulation induced by the characteristics of the nanomaterials. In addition, the phenomenon observed in the treatment with the ZnO nanocrystals corroborates what has been discussed since in this case, parasitemia, amastigote nests and parasite antigens as stimuli for IL-6 and TNF-α were maintained, with consequent maintenance of the levels of these cytokines, weight loss and increased mortality of the animals.

IFN-γ has also been described as being related to tissue damage, cardiomyocyte death and damage to the host, depending on the levels and time of action. However, we believe from our results that the action of this cytokine in the acute phase was focused on eliminating the parasite and not so much on harming the host. This is mainly due to the lack of concomitant participation of TNF-α, which was reduced with the ZnO:9Ag or ZnO:11Ag treatment.

In fact, the use of different concentrations of IFN-γ in vitro has been shown to induce mitochondrial dysfunction in cardiomyocytes. However, when concomitant TNF-α is added, this dysfunction is accentuated [46], which may suggest a deleterious synergistic effect for the host. In addition, NFR1−/− mice treated in the acute phase with a subcurative dose of Benznidazole survived the acute infection and exhibited reduced cardiac inflammation during the chronic phase. In contrast, IFN-γ−/− and IL12−/− mice succumbed soon after the end of treatment with intense parasitemia, parasitism and inflammatory processes [57] Furthermore, in TNFRp55−/− mice, the lack of signaling by this TNF-α receptor was unable to reduce microbicidal mechanisms related to the IFN-γ/NO axis in macrophages in vitro [58]. Thus, based on our results, it is suggested that even without high levels of TNF-α, IFN-γ is able to control, in part, *T. cruzi* infection in the acute phase, which was added to by the trypanocidal action of the ZnO:9Ag and ZnO:11Ag nanomaterials. Thus, the reduction in the number of cardiac amastigote nests in these two treated groups may have been due to the maintenance of IFN-γ levels and the microbicidal activity and trypanocidal capacity of the nanomaterials.

Both in humans [59,60,61] and in experimental models [24,27,62,63,64,65], *T. cruzi* infection has been shown to induce neuronal destruction, especially in the myenteric plexus, whether in the large intestine or the esophagus. This process begins in the acute phase and continues during the chronic phase. In fact, our experimental model follows what is established in the literature, with neuronal reduction in the acute experimental phase. Even so, the groups treated with the highest concentrations of AgO experienced neuronal protection, with the PGP 9.5 labeling observed in the healthy control group re-established. Meanwhile, the ZnO:5Ag nanocomposite, intermediate in terms of the proportions of AgO and pure ZnO nanocrystals, was unable to demonstrate neuroprotection. Thus, it is suggested that two factors may be associated with the neuroprotection: (1) a reduction in ZnO and/or (2) an increase in AgO in the structure. In addition to the reduction in ZnO in the nanomaterials, other parameters may have been related to the neuroprotection, all mediated by the exposure and impact of the ZnO:9Ag or ZnO:11Ag nanocomposites on experimental CD: a reduction in parasitemia, a reduction in the inflammatory infiltrate and a reduction in intestinal TNF-α.

In fact, in the acute phase of the infection, direct parasitism of the enteric neurons and the impact of the anti-*T. cruzi* immune system on the intestines are the main factors in the organ’s denervation, phenomena observed in both in vitro and in vivo models [66,67,68,69]. Thus, this destruction, even in the initial phase of the disease, aids the progression to chronic digestive forms. It has already been shown in models of infection with the Y or Colombian strain that in the acute phase, there is a 40–60% reduction in the neuronal population in the myenteric plexus (colon) of mice, which characterizes intense destruction of these structures in the early stages of infection [24,27,28,66,69]. However, this destruction is not accentuated as the disease progresses to the chronic phase, demonstrating the importance of the acute phase in intestinal denervation. Associated with this intense neuronal destruction, the number of parasites, amount of inflammatory infiltrate and levels of pro-inflammatory cytokines (especially IFN-γ, TNF-α and IL-6) are elevated.

In the context of neurotoxicity, TNF-α, for example, is a pro-inflammatory cytokine produced in the early stages of neuroinflammation and in the chronic stages that participates in the induction of other cytokines in this context [70,71,72,73]. In addition, microglial cells are capable of producing TNF-α, which is also related to the maintenance of neuroinflammation, and this cytokine is capable of inducing neuronal apoptosis when found in high amounts [74,75]. Thus, although the immune system acts to control the infection, it concomitantly induces neuronal destruction, which was remedied by treatment with the ZnO:9Ag or ZnO:11Ag nanocomposites.

Our results show that the use of nanocomposites acted to reduce pro-inflammatory components in the intestines and hearts of a murine model of acute-phase CD. They also show that by reducing pro-inflammatory cytokines, parasitemia and tissue parasitism, the treatments increased the animals’ survival. Although the molecular mechanisms were not evaluated in this work, perspectives are open for understanding how the ZnO:9Ag and ZnO:11Ag nanocomposites modulated the immune response cells in each organ, whether for inflammatory infiltrates in the heart or even microglial cells in the intestine. Our results also open up the prospect of applying these treatments for periods longer than seven days, as well as evaluating their effects in a chronic-phase model of CD.

## 5. Conclusions

This study showed that ZnO:9Ag and ZnO:11Ag nanocomposites acted against Chagas disease in an acute-phase experimental model and thus represent an alternative for testing in the chronic phase. Both treatments were able to reduce parasitemia, reduce the amount of cardiac and intestinal cytokines, such as TNF- and IL-6, and increase the animals’ survival. In addition, treatment with these nanomaterials reduced the amount of intestinal inflammatory infiltrate and prevented neuronal loss in this organ. Thus, it is believed that these nanomaterials may represent an alternative and should be targeted for application in the chronic phase.

## Figures and Tables

**Figure 1 pharmaceutics-16-01415-f001:**
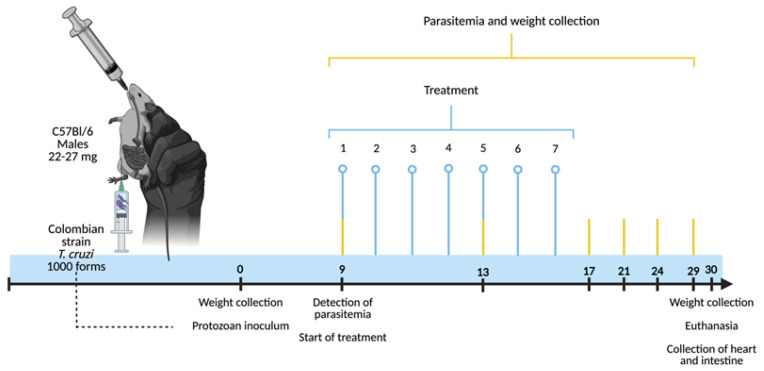
Experimental design for evaluating the potential of silver-doped zinc oxide nanomaterials in the acute experimental phase of Chagas disease. Male C57BL/6 mice, 8 to 10 weeks old and weighing 22–27 g, were infected with 1000 blood trypomastigote forms of the Colombian strain of *T. cruzi* (subcutaneously). On the day of the protozoan inoculation (day 0), the weight of the animals was recorded. The treatments with the ZnO, ZnO:5Ag, ZnO:9Ag or ZnO:11Ag nanomaterials were started concomitantly with the first detection of blood trypomastigote forms (day 9). Thus, by gavage, the treatments were carried out at a dose of 5 mg/kg/d in PBS for seven consecutive days. At the same time, parasitemia, the process of counting circulating trypomastigote forms, took place every four days until the 29th day post-infection. Then, 30 days post-infection, the animals were subjected to collection of data such as their survival and weight, with subsequent euthanasia and collection of their hearts and intestines for histopathological and immunological analyses.

**Figure 2 pharmaceutics-16-01415-f002:**
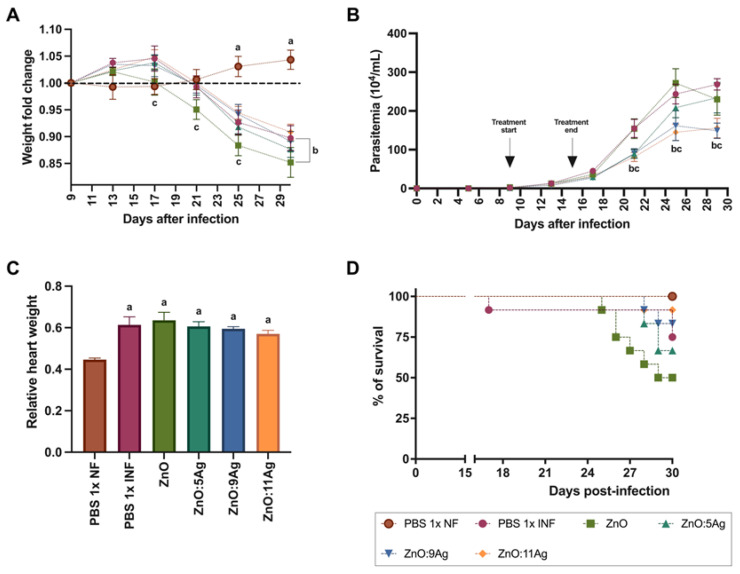
Impact of ZnO, ZnO:5Ag, ZnO:9Ag and ZnO:11Ag nanomaterials on clinical and parasitological parameters of acute *T. cruzi* infection (Colombian strain) in C57Bl/6 mice. (**A**) Change in body weight. (**B**) Parasitemia. (**C**) Relative heart weight. (**D**) % of animal survival. Data obtained from treatment of the mice via gavage for seven consecutive days from the first detection of circulating trypomastigote forms in the blood. Differences were considered statistically significant when *p* < 0.05. ^a^ Difference between treatments with nanomaterials and the healthy group treated with the vehicle (PBS 1×). ^b^ Difference between treatments with nanomaterials and the infected group treated with the vehicle (PBS 1×). ^c^ Difference between treatments with nanomaterials doped with silver oxide and the infected group treated with ZnO nanocrystals. PBS 1× NF: not infected. PBS 1× INF: infected.

**Figure 3 pharmaceutics-16-01415-f003:**
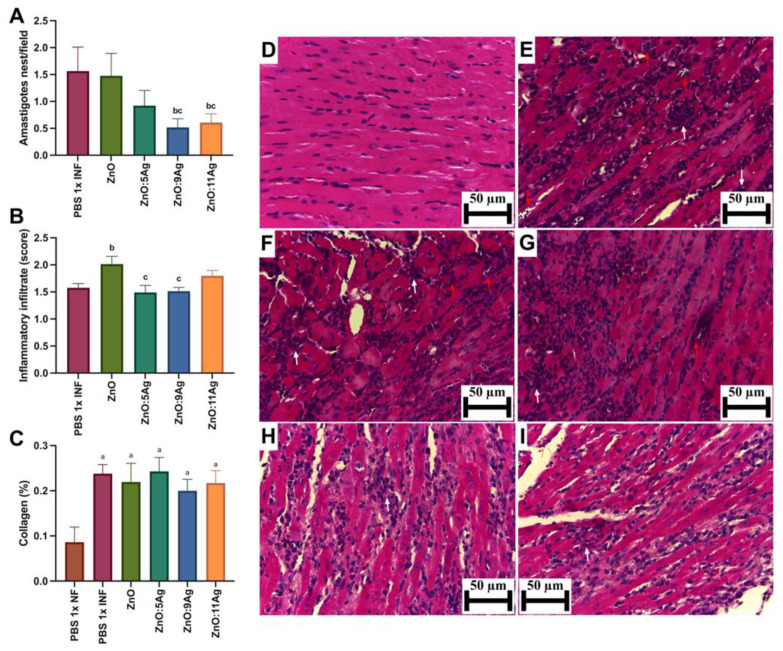
Impact of ZnO, ZnO:5Ag, ZnO:9Ag and ZnO:11Ag nanomaterials on tissue parasitism, cardiac inflammatory infiltrate and collagen deposition in acute (30-day) *T. cruzi* infection (Colombian strain) in C57Bl/6 mice. Quantification of (**A**) amastigote nests (red arrows), (**B**) inflammatory infiltrate (white arrows) and (**C**) collagen (collagen/animal) in the heart. Photomicrographs of heart tissue from (**D**) uninfected mice, (**E**) infected mice treated with vehicle, (**F**) infected mice treated with ZnO, (**G**) infected mice treated with ZnO:5Ag, (**H**) infected mice treated with ZnO:9Ag and (**I**) infected mice treated with ZnO:11Ag. Obtained under an ordinary light microscope with a 40× objective. Material stained with Hematoxylin–Eosin. Data obtained from treatment of the mice via gavage for seven consecutive days from the first detection of circulating trypomastigote forms in the blood. Differences considered statistically significant when *p* < 0.05. ^a^ Difference between treatments with nanomaterials and the healthy group treated with the vehicle (PBS 1×). ^b^ Difference between treatments with nanomaterials and the infected group treated with the vehicle (PBS 1×). ^c^ Difference between treatments with nanomaterials doped with silver oxide and the infected group treated with ZnO nanocrystals. PBS 1× INF: infected.

**Figure 4 pharmaceutics-16-01415-f004:**
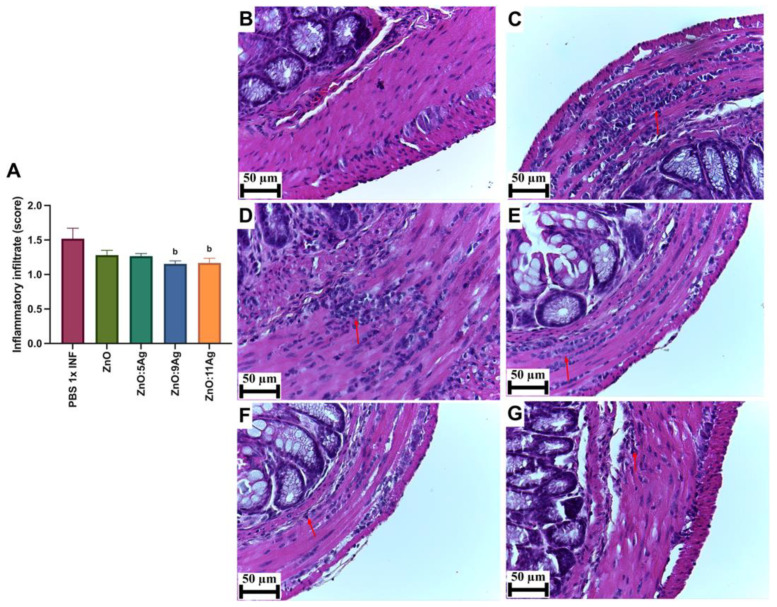
Impact of the nanomaterials ZnO, ZnO:5Ag, ZnO:9Ag and ZnO:11Ag on the intestinal inflammatory infiltrate in acute infection (30 days) with *T. cruzi* (Colombian strain) in C57Bl/6 mice. (**A**) Inflammatory infiltrates in the intestine (red arrow). Photomicrographs of the intestines of (**B**) uninfected mice, (**C**) infected mice treated with the vehicle, (**D**) infected mice treated with ZnO, (**E**) infected mice treated with ZnO:5Ag, (**F**) infected mice treated with ZnO:9Ag and (**G**) infected mice treated with ZnO:11Ag. Obtained under an ordinary light microscope with a 40× objective. Material stained with Hematoxylin–Eosin. Data obtained from treatment of the mice via gavage for seven consecutive days from the first detection of circulating trypomastigote forms in the blood. Differences considered statistically significant when *p* < 0.05. ^b^ Difference between treatments with nanomaterials and the infected group treated with the vehicle (PBS 1×). PBS 1× INF: infected.

**Figure 5 pharmaceutics-16-01415-f005:**
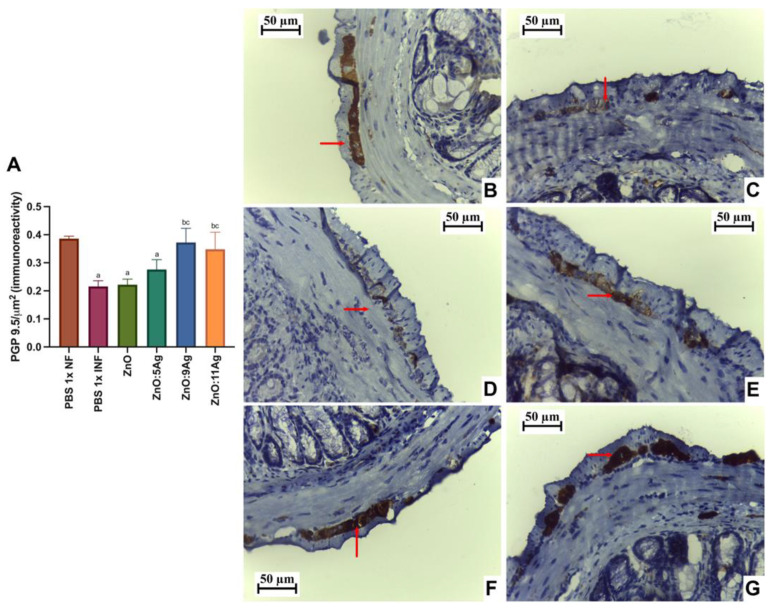
Impact of ZnO, ZnO:5Ag, ZnO:9Ag and ZnO:11Ag nanomaterials on neuronal immunolabeling by intestinal PGP 9.5 (in the myenteric plexus) in acute infection (30 days) with *T. cruzi* (Colombian strain) in C57Bl/6 mice. (**A**) Quantification of immunolabeling for PGP 9.5/µm^2^ (red arrow). Photomicrographs of the immunolabeling in (**B**) the healthy group treated with vehicle, (**C**) the infected group treated with the vehicle and the groups treated with (**D**) ZnO, (**E**) ZnO:5Ag, (**F**) ZnO:9Ag and (**G**) ZnO:11Ag. Obtained under an ordinary light microscope with a 40× objective. Data obtained from treatment of the mice via gavage for seven consecutive days from the first detection of circulating trypomastigote forms in the blood. Differences considered statistically significant when *p* < 0.05. ^a^ Difference between infected groups and the healthy group treated with the vehicle (PBS 1×). ^b^ Difference between treatments with nanomaterials and the infected group treated with the vehicle (PBS 1×). ^c^ Difference between treatments with silver-doped nanomaterials and the infected group treated with ZnO nanocrystals. PBS 1× NF: not infected. PBS 1× INF: infected.

**Figure 6 pharmaceutics-16-01415-f006:**
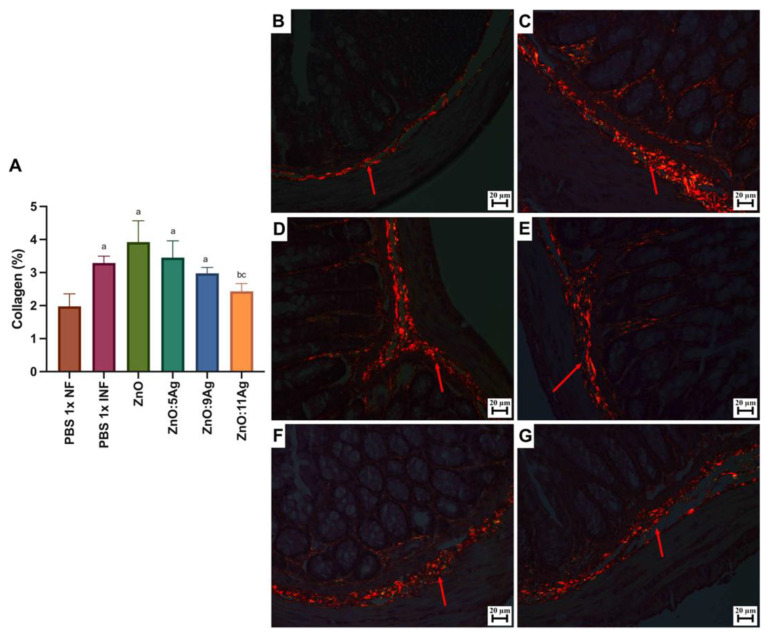
Impact of ZnO, ZnO:5Ag, ZnO:9Ag and ZnO:11Ag nanomaterials on intestinal collagen deposition in acute infection (30 days) with *T. cruzi* (Colombian strain) in C57Bl/6 mice. (**A**) Quantification of collagen in the intestine (collagen/animal) (red arrow). Photomicrographs of intestinal collagen deposition in the (**B**) healthy group treated with the vehicle, (**C**) the infected group treated with the vehicle and groups treated with (**D**) ZnO, (**E**) ZnO:5Ag, (**F**) ZnO:9Ag and (**G**) ZnO:11Ag. Obtained under a polarized light microscope with a 20× objective. Data obtained from treatment of the mice via gavage for seven consecutive days from the first detection of circulating trypomastigote forms in the blood. Differences considered statistically significant when *p* < 0.05. ^a^ Difference between treatments with nanomaterials and the healthy group treated with the vehicle (PBS 1×). ^b^ Difference between treatments with nanomaterials and the infected group treated with the vehicle (PBS 1×). ^c^ Difference between treatments with silver-doped nanomaterials and the infected group treated with the ZnO nanocrystal. PBS 1× NF: not infected. PBS 1× INF: infected.

**Figure 7 pharmaceutics-16-01415-f007:**
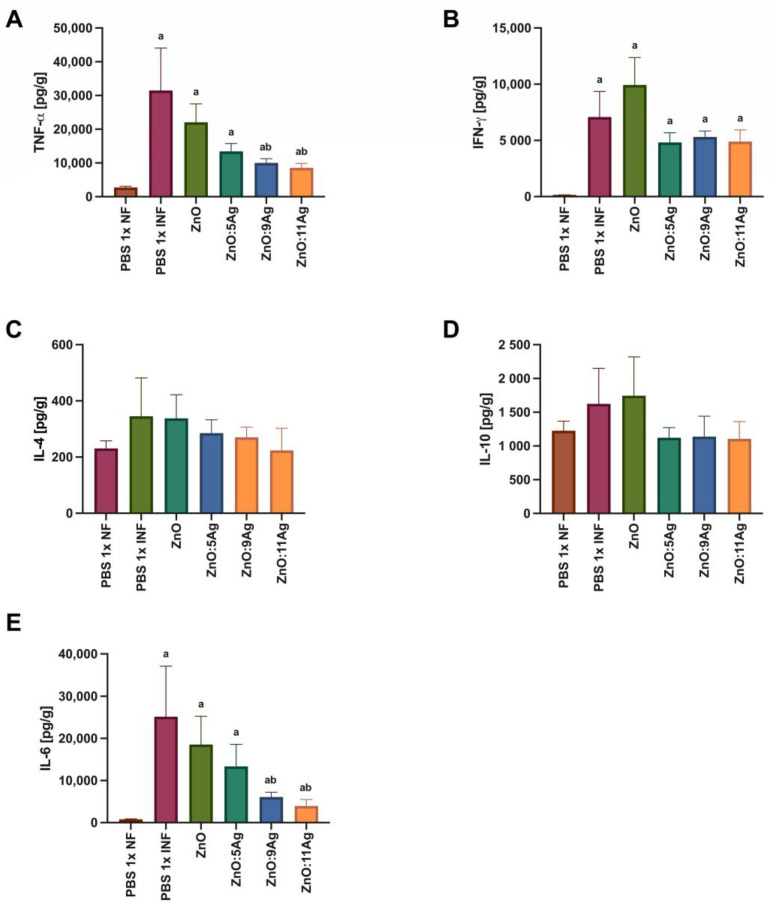
Impact of ZnO, ZnO:5Ag, ZnO:9Ag and ZnO:11Ag nanomaterials on the cardiac cytokine profile in acute infection (30 days) with *T. cruzi* (Colombian strain) in C57Bl/6 mice. Quantification of (**A**) TNF-α, (**B**) IFN-γ, (**C**) IL-4, (**D**) IL-10 and (**E**) IL-6 in the heart in pg of cytokine per g of heart tissue. Data obtained from treatment of the mice via gavage for seven consecutive days from the first detection of circulating trypomastigote forms in the blood. Differences considered statistically significant when *p* < 0.05. ^a^ Difference between treatments with nanomaterials and the healthy group treated with the vehicle (PBS 1×). ^b^ Difference between treatments with nanomaterials and the infected group treated with the vehicle (PBS 1×). PBS 1× NF: not infected. PBS 1× INF: infected.

**Figure 8 pharmaceutics-16-01415-f008:**
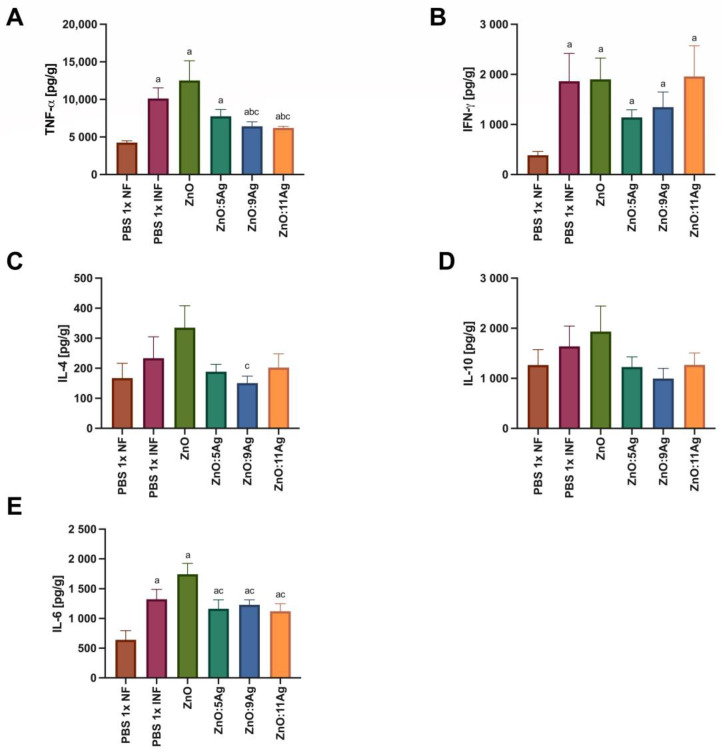
Impact of ZnO, ZnO:5Ag, ZnO:9Ag and ZnO:11Ag nanomaterials on the intestinal cytokine profile in acute infection (30 days) with *T. cruzi* (Colombian strain) in C57Bl/6 mice. Quantification of (**A**) TNF-α, (**B**) IFN-γ, (**C**) IL-4, (**D**) IL-10 and (**E**) IL-6 in the intestine in pg of cytokine per g of intestine. Data obtained from treatment of the mice via gavage for seven consecutive days from the first detection of circulating trypomastigote forms in the blood. Differences considered statistically significant when *p* < 0.05. ^a^ Difference between treatments with nanomaterials and the healthy group treated with the vehicle (PBS 1×). ^b^ Difference between treatments with nanomaterials and the infected group treated with the vehicle (PBS 1×). ^c^ Difference between treatments with silver-doped nanomaterials and the infected group treated with ZnO. PBS 1× NF: not infected. PBS 1× INF: infected.

**Table 1 pharmaceutics-16-01415-t001:** Characterization of ZnO, ZnO:5Ag, ZnO:9Ag and ZnO:11Ag nanomaterials by XDR (X-ray diffraction) and SEM (scanning electronic microscopy).

	Composition (XRD)			Size (SEM)
	ZnO	AgO	Ag-ZnO	
ZnO	100%	-	-	~260 nm
ZnO:5Ag		49%	51%	~250 nm
ZnO:9Ag		65%	35%	~345 nm
ZnO:11Ag		68%	38%	~290 nm

**Table 2 pharmaceutics-16-01415-t002:** Number of groups, *Trypanosoma cruzi* strain used for infection, days of maintenance for each experiment, condition of the animals and number of animals at the end of the experiment.

Groups	*Trypanosoma cruzi* Strain	Condition	Days of Infection/Maintenance	Treatment Dose	Days of Treatment	Route of Administration	Number of Animals
Healthy control (NF)	-	Vehicle	30	-	7 consecutive days	Gavage	6
Infection control (INF)	Colombian	Vehicle	30	-	7 consecutive days	Gavage	12
ZnO ZnO:5Ag ZnO:9Ag ZnO:11Ag	Colombian	Nanomaterial	30	5 mg/kg	7 consecutive days	Gavage	12 12 12 12

**Table 3 pharmaceutics-16-01415-t003:** Summarized results on the impact of treatments with ZnO, ZnO:5Ag, ZnO:9 or ZnO:11Ag in a murine model of acute-phase Chagas disease.

	Nanomaterials			
	ZnO	ZnO:5Ag	ZnO:9Ag	ZnO:11Ag
*Parameters evaluated*				
*Heart*				
Amastigote nests	=	=	↓	↓
Inflammatory infiltrate	↑	=	=	=
Collagen deposition	=	=	=	=
Cytokines	TNF-α: =	TNF-α: =	TNF-α: ↓	TNF-α: ↓
	IFN-γ: =	IFN-γ: =	IFN-γ: =	IFN-γ: =
	IL-6: =	IL-6: =	IL-6: ↓	IL-6: ↓
	IL-4: =	IL-4: =	IL-4: =	IL-4: =
	IL-10: =	IL-10: =	IL-10: =	IL-10: =
*Intestine*				
Inflammatory infiltrate	=	=	↓	↓
Neuronal immunolabeling	=	=	↓	↓
Collagen deposition	=	=	=	↓
Cytokines	TNF-α: =	TNF-α: =	TNF-α: ↓	TNF-α: ↓
	IFN-γ: =	IFN-γ: =	IFN-γ: =	IFN-γ: =
	IL-6: =	IL-6: =	IL-6: =	IL-6: =
	IL-4: =	IL-4: =	IL-4: =	IL-4: =
	IL-10: =	IL-10: =	IL-10: =	IL-10: =
*Clinical and parasitological*				
Weight	↓	=	=	=
Parasitemia	=	=	↓	↓
Survival	↓	=	↑	↑

All the ratios shown here were compared to the infected group treated with the vehicle; thus, = means equal to the infected control; ↑ means increased compared to the infected control; and ↓ means reduced compared to the infected control.

## Data Availability

All the data from this work are available in the article itself and on request from the authors.

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
