# Peer review of "Achieving the Optimal AgO Concentrations to Modulate the Anti-Trypanosoma cruzi Activity of Ag-ZnO/AgO Nanocomposites: In Vivo Investigations"

_pharmaceutics, 2024, doi:10.3390/pharmaceutics16111415_

Round 1
Reviewer 1 Report
Comments and Suggestions for Authors
The paper details a series of experiments examining the effectiveness of zinc and silver-based nanomaterials against Chagas disease (CD) in the C57Bl/6 murine model. The results demonstrated that the more concentrated nanocomposites had significant effects on the parasitemia biomarkers used. The text is well-organized and easy to follow, with graphical evidence clearly illustrating the findings. To enhance the discussion section, it is suggested to include a table summarizing the main authors and their respective results to emphasize their noteworthy findings. Additionally, it would be beneficial to acknowledge the significance of this work in addressing the needs of countries with limited resources in the intertropical and subtropical regions, thereby increasing its social impact
Author Response
Comments 1: The paper details a series of experiments examining the effectiveness of zinc and silver-based nanomaterials against Chagas disease (CD) in the C57Bl/6 murine model. The results demonstrated that the more concentrated nanocomposites had significant effects on the parasitemia biomarkers used. The text is well-organized and easy to follow, with graphical evidence clearly illustrating the findings. To enhance the discussion section, it is suggested to include a table summarizing the main authors and their respective results to emphasize their noteworthy findings. Additionally, it would be beneficial to acknowledge the significance of this work in addressing the needs of countries with limited resources in the intertropical and subtropical regions, thereby increasing its social impact.
Response 1: Thank you for your comments. As suggested, we have added a table summarizing the results (see lines 492-498).
Reviewer 2 Report
Comments and Suggestions for Authors
General comments
The authors claims that the goal of the study was to evaluate the in vivo trypanocidal effect of ZnO nanocrystals and ZnO/AgO nanocomposites using an experimental model of the acute phase of Chagas Diseases (CD). For that, ZnO nanocrystals and ZnO/AgO nanocomposites with different proportions of silver (ZnO:5Ag, ZnO:9Ag and ZnO:11Ag) was administrated in C57Bl/6 mice and the infection was done with the Colombian strain of T. cruzi. The weight, parasitemia and survival were assessed during treatment and up to the day of euthanasia. The cardiac and intestinal parasitism, inflammatory infiltrate, collagen deposition and cytokine quantification were analyzed after euthanasia.
Although it is very important to discovery and development new alternative treatments to CD, the proposed work has several methodological issues and the results could be questionable.
Although it is very important to discovery and development new alternative treatments to CD, the proposed work has several methodological issues and the results could be questionable.
The methods described in “Processing of samples for histopathology" (2.5.), “Cardic Tissue parasitism” (2.5.1.), “Quantification of cardiac and intestinal inflammatory infiltrate” (2.5.2.), “Quantification of PGP 9.5 immunoreactivity in the myenteric plexus” (2.5.3.), and “Quantification of cardiac and intestinal collagen” (2.5.4.) needs references that support (validate) the methods used.
All these methods were based on microscopic observation of tissues after staining (10 fields/mouse or 30 fields/mouse using x10, x20 or 40x objectives). The methods depend on the microscope model, specimen preparation methods and on the operator skills. The errors in microscopy images and in interpretation can lead to false conclusions. Therefore, is crucial the use of validate methods (previously or actual) to confirm the results, to avoid bias and sustained the discussion and conclusion (see Jost AP, Waters JC. Designing a rigorous microscopy experiment: Validating methods and avoiding bias. J Cell Biol. 2019 May 6;218(5):1452-1466. doi: 10.1083/jcb.201812109).
Specific comments
Lines 23-24: Ag and AgO?
Line 36: Ag/ AgO is correct?
Lines 51- 61: the authors highlight the treatment limitation of Chagas Diseases, especially in Chronic phase (CD) and the urgent need to development/discovery new compounds. However, the work was done using a model of the acute phase of the disease. Why this choose?
Lines 67-68: improve English of these two sentences.
Line 79: And what is happens in host cells? what is the toxicity of this nanocrystals? This information should be introducing at this point in the introduction.
Line 81: What is AgO? the study evaluated the activity of silver (Ag) and silver oxide (AgO) in ZnO nanocomposites? Please standardize the abbreviations/acronyms used for nanocomposites.
Lines 89-94: the work evaluated ZnO and ZnO/AgO? or ZnO and ZnO/Ag/AgO? or Zn and ZnO/Ag? the use of different abrevations along the text make confunsing. Please standardize the abbreviations/acronyms used for nanocomposites.
Line 119: How much animals per group?
Line 122: How much animals per treated group?
Line 125: the legend of table 2 should be improved.
Table 2: Suggestion: add to "Strain" the species identification of " Trypanosoma cruzi"
Line 132: After the infection (inoculation of trypomastigotes), how long does it take until detection of parasitemia? this time was equal in all animals?
Line 141: Only for cytokine quantification?
Figure 1: In the image of experimental design until day 0 (inoculation of trypomastigotes) the experience should not be named by "Days of infection" (painting blue color).
Lines 170 - 177: in 2.5. "Processing of samples for histopathology" needs reference (s) that support the method.
Lines 180 - 186: in 2.5.1. "Cardiac tissue parasitism needs reference (s) that support the method.
Lines 188 - 198: in 2.5.2 "Quantification of cardiac and intestinal inflammatory infiltrate" needs reference (s) that support the method.
Lines 204-217: in 2.5.3. " Quantification of PGP 9.5 immunoractivity in the myenteris plexus" needs reference (s) that support the method.
Lines 220-226: in 2.5.4."Quantification of cardiac and intestinal collagen" needs reference (s) that support the method.
Line 284: ZnO:5Ag and ZnO seems very toxic to animals. Its kill more animals that the infection by T. cruzi. Previously the team had evaluated whether the amount of Ag in ZnO NPs would interfere with the toxicity of nanomaterials (do Carmo Neto et al., 2024) and showed that the dose of 5 mg/kg/d for all nanomaterials could be considered safe and of low toxicity. However, at present study it seems not true (evaluating % of Survival).
Figure 2:
i) the use of different symbols and different colors for each treatment conditions make difficulties to analysis the graphics. Suggestion: using different colors for each treatment conditions (and only one symbol type); or different symbols for each treatment conditions (and black color for all);
ii) The groups PBS 1x NF and PBS 1x INF should be identified in the legend;
iii) The representation of statistical analysis is confused and the use of “a. b. c.” is not adequate. Suggestion: insert the statistical analysis in the graphs by using symbols (#, &, ...);
iv) In graphic B what means "bcd"? "d." of what?
Lines 312-314: the elevated number of Inflammatory cells could be due to the high toxicity observed in the assay of % of Survival.
Figure 3:
i) The photos should include the identification of the inflammatory infiltrate by a symbol ou arrows. Moreover, the light of each image is very different; so the authors should inclued in legend the DPIs (Dots Per Inch) of each image (C to H).
ii) The amplification of each image should be identify by bars in each image; is not enough put in the figure legend the use of the 40x objective .
iii) The representation of statistical analysis is confused and the use of “ b. c.” is not adequate. Suggestion: insert the statistical analysis in the graphs by using symbols (#, &,..);
iv) In graphic B what means "bcd"? "d." of what?
Figure 4:
i) The representation of statistical analysis is confused and the use of "b. c." is not adequate. Suggestion: insert the statistical analysis in the graphs by using symbols ( #, &, ..);
ii) The photos present poor quality; it impossible to see anything.
Figure 5:
i) The photos should included the identification of the inflammatory infiltrate by a symbol ou arrows.
ii) The amplification of each image shoud be identify by bars in each image; is not enouph put in the figure legend the use of 40x objective.
iii) The representation of statistical analysis is confused and the use of "b." is not adequate. Sugesttion: insert the statistical analysis in the graphs by another way with symbols (#, &).
Figure 6:
i) All photos should include the identification of the PGP 9.5 by arrows (like in the image B).
ii) The amplification should be identifying by bars in each image; is not enough put in the figure legend the use of 40x objective.
iii) The representation of statistical analysis is confused and the use of "a. b. " is not adequate. Suggestion: insert the statistical analysis in the graphs by symbols (#, &).
Figure 7:
i) the amplification shoud be identiy by bars in each image; is not enouph put in the figure legend the use of 20x objective.
ii) The representation of statistical analysis is confused and the use of "a. b. " is not adequate. Suggestion: insert the statistical analysis in the graphs by symbols (#, &).
Figure 8: The representation of statistical analysis is confused and the use of "a. b. " is not adequate.. Suggestion: insert the statistical analysis in the graphs by symbols (#, &).
Figure 9: The representation of statistical analysis is confused and the use of "a. b. c." is not adequate. Suggestion: insert the statistical analysis in the graphs by symbols (#, &).
Line 483: change "(PBS 1x)" to "(PBS 1x NF)"
Line 484: change "(PBS 1x)" to "(PBS 1x INF)"
Lines 505-506: Please change the nanocomposites designations because are not equal that of Table 1: 51% ZnO and 49% AgO, etc. It is very important using the same designation/abbreviation for the formulations along all the text, tables and figures.
Lines 509-511: revise the English.
Lines 568-577: revise the English in the sentences.
Lines 643-648: revise the English in the sentences.
References: the authors should reduce the number of references doing a selection for that are essential to support and discuss the present work. Multiple references exploring similar matters.

The authors should revise several sentences in the manuscript to improve the English.
Author Response
Reviewer #2:
The authors claims that the goal of the study was to evaluate the in vivo trypanocidal effect of ZnO nanocrystals and ZnO/AgO nanocomposites using an experimental model of the acute phase of Chagas Diseases (CD). For that, ZnO nanocrystals and ZnO/AgO nanocomposites with different proportions of silver (ZnO:5Ag, ZnO:9Ag and ZnO:11Ag) was administrated in C57Bl/6 mice and the infection was done with the Colombian strain of T. cruzi. The weight, parasitemia and survival were assessed during treatment and up to the day of euthanasia. The cardiac and intestinal parasitism, inflammatory infiltrate, collagen deposition and cytokine quantification were analyzed after euthanasia.
Authors’ answer:
Thank you very much for your comments. Below are all the answers to the reviewer.
Comments 1:
The methods described in “Processing of samples for histopathology" (2.5.), “Cardic Tissue parasitism” (2.5.1.), “Quantification of cardiac and intestinal inflammatory infiltrate” (2.5.2.), “Quantification of PGP 9.5 immunoreactivity in the myenteric plexus” (2.5.3.), and “Quantification of cardiac and intestinal collagen” (2.5.4.) needs references that support (validate) the methods used.
All these methods were based on microscopic observation of tissues after staining (10 fields/mouse or 30 fields/mouse using x10, x20 or 40x objectives). The methods depend on the microscope model, specimen preparation methods and on the operator skills. The errors in microscopy images and in interpretation can lead to false conclusions. Therefore, is crucial the use of validate methods (previously or actual) to confirm the results, to avoid bias and sustained the discussion and conclusion (see Jost AP, Waters JC. Designing a rigorous microscopy experiment: Validating methods and avoiding bias. J Cell Biol. 2019 May 6;218(5):1452-1466. doi: 10.1083/jcb.201812109).
Response 1:
Thank you very much for your comment. As suggested, we have added references that have already validated the same methods used in this work (see “Processing of samples for histopathology"-2.5.; “Cardic Tissue parasitism”-2.5.1.; “Quantification of cardiac and intestinal inflammatory infiltrate”-2.5.2.; “Quantification of PGP 9.5 immunoreactivity in the myenteric plexus”-2.5.3.; and “Quantification of cardiac and intestinal collagen”-2.5.4. In addition, our team has different papers that use the same methods that are very well accepted in the literature. This also proves our experience, which further reduces the chance of analysis errors.
Comments 2:
Lines 23-24: Ag and AgO?
Line 36: Ag/ AgO is correct?
Response 2:
Thank you for your questions. In lines 23-25, the correct text is AgO. The nanocomposites used in this work are products of the combination of ZnO nanocrystals and AgO nanoparticles. Thus, as described in the methodology (2.1. Nanomaterials) and shown in Table 1 (see lines 88-101), the materials have a certain % of ZnO, a certain % of AgO and a certain % of Ag.
This results in the following nanomaterials:
ZnO = 100% ZnO, 0% AgO; 0% Ag
ZnO:5Ag = 51% ZnO, 49% AgO, 5%Ag
ZnO:9Ag = 35% ZnO, 65% AgO, 9% Ag
ZnO:11Ag = 38% ZnO, 68% AgO, 11% Ag
Comments 3:
Lines 51- 61: the authors highlight the treatment limitation of Chagas Diseases, especially in Chronic phase (CD) and the urgent need to development/discovery new compounds. However, the work was done using a model of the acute phase of the disease. Why this choose?
Lines 67-68: improve English of these two sentences.
Response 3:
Thank you very much for your comments. To develop a new treatment, it is necessary to carry out a screening to assess the possible effect of the components on the experimental disease. Thus, using an acute phase model first ensures that the treatment in question has the potential to go on to chronic/undetermined phase trials. In the acute phase, using only a few animals and a relatively short period of time allows the potential of the new compound to be assessed, above all through the survival of the animals. Chronic phase models for Chagas disease require many animals, especially when infected with the Colombian strain, which induces high mortality and represents a certain resistance in animal ethics committees. As soon as the new component is seen to be effective in the acute phase, it can be moved on to the chronic phase.
Comments 4:
Line 79: And what is happens in host cells? what is the toxicity of this nanocrystals? This information should be introducing at this point in the introduction.
Line 81: What is AgO? the study evaluated the activity of silver (Ag) and silver oxide (AgO) in ZnO nanocomposites? Please standardize the abbreviations/acronyms used for nanocomposites.
Lines 89-94: the work evaluated ZnO and ZnO/AgO? or ZnO and ZnO/Ag/AgO? or Zn and ZnO/Ag? the use of different abrevations along the text make confunsing. Please standardize the abbreviations/acronyms used for nanocomposites.
Response 4:
Thank you very much for the comments. The names of the nanomaterials are standardized throughout the text. As explained in the methodology, the nanomaterials are made of ZnO and AgO and contain a certain % of Ag (Table 1).
Comments 5:
Line 119: How much animals per group?
Line 122: How much animals per treated group?
Response 5:
Thank you for your questions. For the group of uninfected animals, 6 animals were used. For the treated or untreated infected groups, 12 were used. This information has been added to Table 2 (see lines 125-126).
Comments 6:
Line 125: the legend of table 2 should be improved.
Table 2: Suggestion: add to "Strain" the species identification of " Trypanosoma cruzi"
Line 132: After the infection (inoculation of trypomastigotes), how long does it take until detection of parasitemia? this time was equal in all animals?
Line 141: Only for cytokine quantification?
Response 6:
Thank you very much for your comments.As suggested, we have added “Trypanosoma cruzi strain” in the table legend, as well as in the body of the table.
After infection with the Colombian strain, it takes 9 days for the first parasites to be detected in the bloodstream. All the mice showed parasitemia at 9 days post-infection. Line 141: The frozen tissue was only used for cytokine measurement.
Comments 7:
Figure 1: In the image of experimental design until day 0 (inoculation of trypomastigotes) the experience should not be named by "Days of infection" (painting blue color).
Lines 170 - 177: in 2.5. "Processing of samples for histopathology" needs reference (s) that support the method.
Lines 180 - 186: in 2.5.1. "Cardiac tissue parasitism needs reference (s) that support the method.
Lines 188 - 198: in 2.5.2 "Quantification of cardiac and intestinal inflammatory infiltrate" needs reference (s) that support the method.
Lines 204-217: in 2.5.3. " Quantification of PGP 9.5 immunoractivity in the myenteris plexus" needs reference (s) that support the method.
Lines 220-226: in 2.5.4."Quantification of cardiac and intestinal collagen" needs reference (s) that support the method.
Line 284: ZnO:5Ag and ZnO seems very toxic to animals. Its kill more animals that the infection by T. cruzi. Previously the team had evaluated whether the amount of Ag in ZnO NPs would interfere with the toxicity of nanomaterials (do Carmo Neto et al., 2024) and showed that the dose of 5 mg/kg/d for all nanomaterials could be considered safe and of low toxicity. However, at present study it seems not true (evaluating % of Survival).
Response 7:
Thank you very much for your comments. As suggested, we have chosen to exclude “Days of infection” from the figure. Other references supporting the microscopy evaluation methods used in our work have been added (see methodology - topics 2.5-2.5.4).
Line 284: In fact, in the article “Toxicity Assessment of New Ag-ZnO/AgO Nanocomposites: An In Vitro and In Vivo Approach” published by our group, it was shown that a dose of 5 mg/kg/d was not considered toxic for any of the nanomaterials. However, in this context, the animals were not infected. Therefore, it is not possible to make this comparison. We believe that the zinc oxide present in the nanomaterial may somehow influence the worsening of the condition when infection is present. In fact, this phenomenon can be observed in part by the increase in the inflammatory infiltrate and cytokines in the hearts of infected animals treated with ZnO. This discussion is included in detail throughout the article.
Comments 8:
Figure 2:
- i) the use of different symbols and different colors for each treatment conditions make difficulties to analysis the graphics. Suggestion: using different colors for each treatment conditions (and only one symbol type); or different symbols for each treatment conditions (and black color for all);
- ii) The groups PBS 1x NF and PBS 1x INF should be identified in the legend;
iii) The representation of statistical analysis is confused and the use of “a. b. c.” is not adequate. Suggestion: insert the statistical analysis in the graphs by using symbols (#, &, ...);
- iv) In graphic B what means "bcd"? "d." of what?
Lines 312-314: the elevated number of Inflammatory cells could be due to the high toxicity observed in the assay of % of Survival.
Figure 8: The representation of statistical analysis is confused and the use of "a. b. " is not adequate.. Suggestion: insert the statistical analysis in the graphs by symbols (#, &).
Figure 9: The representation of statistical analysis is confused and the use of "a. b. c." is not adequate. Suggestion: insert the statistical analysis in the graphs by symbols (#, &).
Response 8:
Thank you very much for your comments. We believe that the way the results are presented makes it easier to identify the groups and their statistical differences due to the amount of data presented in the article. In addition, we have always used the same colors for the same groups, as well as the same letters a,b,c to elucidate the same statistical differences when found. In addition to the methodology text, the respective figure legends also have the appropriate description, exemplifying the meaning of the letters that are kept from the beginning to the end of the article. All the groups are duly identified in the captions, as well as the text of the results. To improve understanding, PBS 1x NF: not infected. PBS 1x INF: infected at the end of all legends to highlight the uninfected group from the infected group.
In figure 2, graph B, we noticed that d was added incorrectly. We have therefore deleted the letter from the figure.
Comments 9:
Figure 3:
- i) The photos should include the identification of the inflammatory infiltrate by a symbol ou arrows. Moreover, the light of each image is very different; so the authors should inclued in legend the DPIs (Dots Per Inch) of each image (C to H).
- ii) The amplification of each image should be identify by bars in each image; is not enough put in the figure legend the use of the 40x objective .
iii) The representation of statistical analysis is confused and the use of “ b. c.” is not adequate. Suggestion: insert the statistical analysis in the graphs by using symbols (#, &,..);
- iv) In graphic B what means "bcd"? "d." of what?
Response 9:
Thank you very much for your comments. As suggested, we have included the scale bar in all the figures. In addition, we have highlighted the inflammatory infiltrate and the amastigote nests in the figures. We have highlighted the respective colors of the arrows in the legend: red arrow for amastigote nest and white arrow for inflammatory infiltrate (see lines 320-321).
Comments 10:
Figure 4:
- i) The representation of statistical analysis is confused and the use of "b. c." is not adequate. Suggestion: insert the statistical analysis in the graphs by using symbols ( #, &, ..);
- ii) The photos present poor quality; it impossible to see anything.
Response 10:
Thank you very much for your comments. We have enlarged the pictures as well as generating them in the maximum quality allowed in tiff format (1200 dpi).
Comments 11:
Figure 5:
- i) The photos should included the identification of the inflammatory infiltrate by a symbol ou arrows.
- ii) The amplification of each image shoud be identify by bars in each image; is not enouph put in the figure legend the use of 40x objective.
iii) The representation of statistical analysis is confused and the use of "b." is not adequate. Sugesttion: insert the statistical analysis in the graphs by another way with symbols (#, &).
Response 11:
Thank you very much for your comments. As suggested, we have included the scale bar in all the figures and identified the inflammatory infiltrate with red arrows. We have also highlighted the appropriate identification in the legend (see line 365).
Comments 12:
Figure 6:
- i) All photos should include the identification of the PGP 9.5 by arrows (like in the image B).
- ii) The amplification should be identifying by bars in each image; is not enough put in the figure legend the use of 40x objective.
iii) The representation of statistical analysis is confused and the use of "a. b. " is not adequate. Suggestion: insert the statistical analysis in the graphs by symbols (#, &).
Response 12:
As suggested, we have included the scale bar in all the figures and identified the marking for PGP 9.5 in all the figures with red arrows. We have also highlighted the appropriate identification in the legend (see lines 389-390).
Comments 13:
Figure 7:
- i) the amplification shoud be identiy by bars in each image; is not enouph put in the figure legend the use of 20x objective.
- ii) The representation of statistical analysis is confused and the use of "a. b. " is not adequate. Suggestion: insert the statistical analysis in the graphs by symbols (#, &).
Response 13:
Thank you very much for your comments. As suggested, we have included the scale bar in all the figures and identified the collagen fibers with red arrows. We have also highlighted the appropriate identification in the legend (see line 411-412).
Comments 14:
Line 483: change "(PBS 1x)" to "(PBS 1x NF)"
Line 484: change "(PBS 1x)" to "(PBS 1x INF)"
Lines 505-506: Please change the nanocomposites designations because are not equal that of Table 1: 51% ZnO and 49% AgO, etc. It is very important using the same designation/abbreviation for the formulations along all the text, tables and figures.
Lines 509-511: revise the English.
Lines 568-577: revise the English in the sentences.
Lines 643-648: revise the English in the sentences.
References: the authors should reduce the number of references doing a selection for that are essential to support and discuss the present work. Multiple references exploring similar matters.
Response 14:
Thank you very much for your comments. We have made the suggested changes in full. Regarding the references, we believe that all those cited are important for a complete understanding of the results and the discussions suggested. Currently, the trend in some journals is to reduce the number of citations. However, this reduction means that the information is lost and even makes it difficult for the reader to find the basic experiments that confirm or partially discuss the results presented. We believe that the references we have cited contribute to an understanding of the impact of nanoparticles on the parasite and on the immune response and thus help the reader to better understand the results. In addition, we have included more references in relation to the methods that were used.
Reviewer 3 Report
Comments and Suggestions for Authors
In the present manuscript the authors present the in vivo investigations for achieving the optimal AgO of ZnO/AgO nanocomposites to modulate the anti- 2 Trypanosoma cruzi activity. The main question addressed by this research is the evaluation in vivo of ZnO nanocrystals and ZnO/AgO nanocomposites, mainly on clinical and parasitological parameters, as a new therapeutic approach of these materials to combat the infection caused by T. cruzi. The topic is of interest as new alternatives that are less toxic and more selective for the protozoan are needed. The studied topic is relevant to the field of the development of new pharmaceuticals for chronic diseases and addresses important new data for the possible use of metallic and semiconductor nanoparticles for this purpose. The methodology is well designed and implemented and the presentation of the work is very well done. My only concern comes from the fact that these materials have been already published concerning their characterization and their consideration on their leishmanicidal and immunomodulatory activity. But as these studies were in vitro the new in vivo studies add new important information to the subject area. The conclusions address the main question of the research which is the possible use of these nanomaterials in the chronic phase of the studied disease. The references are appropriate.and the Figures are clear.
So, I recommend publication of this article in its present form.
Comments on the Quality of English LanguageThe quality of English is very satisfactory.
Author Response
Comments 1:
In the present manuscript the authors present the in vivo investigations for achieving the optimal AgO of ZnO/AgO nanocomposites to modulate the anti-Trypanosoma cruzi activity. The main question addressed by this research is the evaluation in vivo of ZnO nanocrystals and ZnO/AgO nanocomposites, mainly on clinical and parasitological parameters, as a new therapeutic approach of these materials to combat the infection caused by T. cruzi. The topic is of interest as new alternatives that are less toxic and more selective for the protozoan are needed. The studied topic is relevant to the field of the development of new pharmaceuticals for chronic diseases and addresses important new data for the possible use of metallic and semiconductor nanoparticles for this purpose. The methodology is well designed and implemented and the presentation of the work is very well done. My only concern comes from the fact that these materials have been already published concerning their characterization and their consideration on their leishmanicidal and immunomodulatory activity. But as these studies were in vitro the new in vivo studies add new important information to the subject area. The conclusions address the main question of the research which is the possible use of these nanomaterials in the chronic phase of the studied disease. The references are appropriate and the Figures are clear.
So, I recommend publication of this article in its present form.
Response 1:
Thank you very much for your comments. Although the characterization of nanomaterials has already been published in other papers, our results here are undetected and applied to the in vivo model of Chagas disease. In addition, we have chosen to include the characterization results in this manuscript to help the reader get a more complete picture of how nanomaterials can work. At the same time as including these results in this work, we have highlighted in the methodology the references from which the nanomaterials were characterized.